# Feasibility and readiness to implement Robson classification to monitor caesarean sections in public hospitals in Myanmar: Formative research

Kyaw Lwin Show[1,2]*, Thae Maung Maung[1], Aung Pyae Phyo[1], Kyaw Thet Aung[1], Chetta Ngamjarus[3], Nyein Su Aye[4], Özge Tunçalp[5], Ana Pilar Betrán[5], Saw Kler Ku[6], Pisake Lumbiganon[7], Khaing Nwe Tin[8], Nwe Oo Mon[9], Meghan A. Bohren[10]

1 Department of Medical Research, Ministry of Health, Yangon, Myanmar, 2 Department of Epidemiology and Biostatistics Department, Doctor of Epidemiology and Biostatistics Program, Khon Kaen University, Khon Kaen, Thailand, 3 Faculty of Public Health, Department of Epidemiology and Biostatistics, Khon Kaen University, Khon Kaen, Thailand, 4 Independent Expert, Yangon, Myanmar, 5 Department of Sexual and Reproductive Health and Research, UNDP/UNFPA/UNICEF/WHO/World Bank Special Programme of Research, Development and Research Training in Human Reproduction (HRP), World Health Organization, Geneva, Switzerland, 6 Department of Obstetrics and Gynaecology, University of Medicine 2, Yangon, Myanmar, 7 Faculty of Medicine, Department of Obstetrics and Gynaecology, Khon Kaen University, Khon Kaen, Thailand, 8 Department of Public Health, Maternal and Reproductive Health Division, Ministry of Health, Naypyitaw, Myanmar, 9 Nassau University Medical Center, New York, New York, United States of America, 10 Gender and Women's Health Unit, Centre for Health Equity, School of Population and Global Health, University of Melbourne, Melbourne, Victoria, Australia

* kyawlwins@gmail.com

**Data Availability Statement:** The data/transcripts are not publicly available due to the need to protect participant confidentiality but are available on

## Abstract

Recent years have demonstrated an increase in caesarean section (CS) in most countries worldwide with considerable concern for the potential consequences. In 2015, WHO proposed the use of Robson classification as a global standard for assessing, monitoring and comparing CS rates. Currently, there is no standardized method to assess CS in Myanmar. The aim of this study was to explore health provider's perceptions about the feasibility, acceptability and readiness to implement the Robson classification in public hospitals across Myanmar. Ten maternities were purposively chosen, including all five teaching hospitals (tertiary referral hospital-level) affiliated to each medical university in Myanmar, which provide maternal and newborn care services, and district/township hospitals. Face-to-face in-depth interviews (IDI) with healthcare providers and facility administrators were conducted using semi-structured discussion guides. Facility and medical records systems were also assessed. We used the thematic analysis approach and Atlas.ti qualitative analysis software. A total of 67 IDIs were conducted. Most participants had willingness to implement Robson classification if there were sufficient human resources and training. Limited human resources, heavy workloads, and infrastructure resources were the major challenges described that may hinder implementation. The focal person for data entry, analysis, or reporting could be differed according to the level of facility, availability of human resources, and ability to understand medical terms and statistics. The respondents mentioned the

reasonable request. Requests can be directed to the corresponding author or Institutional Review Board at Department of Medical Research, Myanmar via email (ercdmr2015@gmail.com) or website (https://dmr.gov.mm).

**Funding:** TMM received funding from UNDP/UNFPA/UNICEF/WHO/World Bank Special Programme of Research, Development and Research Training in Human Reproduction (HRP), Department of Sexual and Reproductive Health and Research, WHO to conduct the study. KLS received funding from the HRP Alliance, part of the UNDP-UNFPA-UNICEF-WHO-World Bank Special Programme of Research, Development and Research Training in Human Reproduction (HRP), a co-sponsored programme executed by the World Health Organization (WHO), to complete his doctoral studies. MAB's time is supported by an Australian Research Council Discovery Early Career Researcher Award (DE200100264) and a Dame Kate Campbell Fellowship. This article represents the views of the named authors only and does not represent the views of the World Health Organization.

**Competing interests:** The authors have declared that no competing interests exist.

important role of policy enforcement for the sustainability of data collection, interpretation and feedback. The optimal review interval period could therefore differ according to the availability of responsible persons, and the number of births. However, setting a fixed schedule according to the specific hospital for continuous monitoring of CS rate is required. In Myanmar, implementation of Robson classification is feasible while key barriers mainly related to human resource and training must be addressed to sustain.

## Introduction

Recent years have demonstrated an increase in caesarean section (CS) in many countries, but the reasons for this trend are not well understood. Increasing CS rates are a public health concern due to increased maternal and perinatal risks, cost issues, healthcare efficiency, and inequities [1,2]. In order to better understand contributing factors for the increasing trends and implement measures to appropriately reduce or increase CS, tools are needed to monitor CS, processes and outcomes in a standardized, meaningful and action-oriented manner [3]. A WHO-led systematic review [4] concluded that the Robson ten-group classification system [5] was the best classification system to apply internationally due to its simplicity, clinically-relevant data, accountability, replicability, verifiability, and woman-centeredness. In 2015, WHO proposed the use of Robson classification as a global standard for assessing, monitoring and comparing CS rates both within healthcare facilities and between them [3]. The Robson classification is a system that categorizes all women giving birth in a specific setting into one of 10 groups [3]. Each of the ten groups are mutually exclusive and totally inclusive, meaning that every woman admitted for childbirth can be immediately classified in one (and only one) group based on a few obstetrical characteristics (parity, previous scar, onset of labour, number of fetuses, gestational age, fetal presentation) that are typically collected in healthcare facilities worldwide [3]. The Robson classification can be used to monitor CS rates and outcomes over time in more homogeneous groups of women, identify groups where interventions may be needed to increase or decrease the use of CS, and to develop recommendations for improving clinical management and quality of maternal care [3].

Myanmar has made progress in improving maternal health, with a 60.7% reduction in the maternal mortality ratio (MMR) between 1990 and 2015 [6]. The most recent Myanmar Demographic and Health Survey (2015–16) reported that 37% of births in the 5 years preceding the survey occurred in health facilities, with a population-level CS rate of 17.1% among all live births [7]. Within public health facilities, 43% of births were by CS, compared to 65% of births by CS in private health facilities, noting that based on most recent estimates, approximately two-thirds of women in Myanmar gave birth at home [7]. There are important within-country inequalities in CS in Myanmar. Across all regions, CS rates at a population-level range from 6.1% in Chin state to 25.8% in Yangon region, women in the wealthiest quintile are nine times as likely to give birth by CS than women in the poorest quintile, and women with more than secondary education are 13 times more likely to give birth by CS compared to women with no education [7].

Moreover, there is currently no standardized method to analyze CS practices and outcomes at the facility-level in Myanmar, thus limiting what can be done to understand where and in what populations of women CS is underused or overused. Implementing a classification system is therefore a first step to understand the use of CS in Myanmar, before identifying and implementing interventions to optimize its use. The aim of this study was to explore health provider's perceptions about the feasibility, acceptability and readiness to implement the Robson classification system in public hospitals across Myanmar.

## Materials and methods

### Ethics statement

This study was approved by the Institutional Review Board, Department of Medical Research (Approval number–Ethics/DMR/2019/031), scientific and ethics approval were received from the Review Panel on Research Projects (RP2) at the UNDP/UNFPA/UNICEF/WHO/World Bank Special Programme of Research, Development and Research Training in Human Reproduction, Department of Sexual and Reproductive Health at WHO, the WHO Ethical Review Committee (A65935), and Center for Ethics in Human Research, Khon Kaen University (HE642306).

After informing potential participants about the study, the participants were invited to take part in the study and asked for the written informed consent. All the face-to-face interviews took place in a private room in the study health facilities, such as the break room.

### Study design, context and study sites

This was a formative study using a qualitative data collection and readiness assessment by observation checklist. This analysis is reported according to the Consolidated criteria for reporting qualitative research (COREQ) (S1 Checklist) [8].

In Myanmar, hospitals and maternal and child health clinics (MCH) provide maternal health services in urban areas while station hospitals (sub-township level), rural health centers (RHCs) and sub-health centers provide services in rural areas. Station hospitals are the lowest entry of care where patients receive care from medical doctors. Obstetrics and Gynecology specialist doctors are available at the district hospitals and higher level. The medical doctor to population ratio is 0.7 per 1000 population in 2018 (for comparison, the ratio is 1.7 per 1000 in Thailand and 3.8 per 1000 in Europe) [9]. There are five medical universities in Myanmar: University of Medicine 1 (Yangon), University of Medicine 2 (Yangon), University of Medicine (Mandalay), University of Medicine (Magway), and University of Medicine (Taunggyi).

Ten study sites were purposively chosen, including all five teaching hospitals (tertiary referral hospital level) affiliated to each medical university which provide maternal and newborn care services, and one district or township hospital each from the same Region or State where the selected teaching hospitals are located. All the study hospitals were using paper-based medical records systems at the time of data collection and none was using the Robson Classification.

### Study participants, recruitment and sampling

Two groups of study participants were recruited for in-depth interviews (IDI): healthcare providers (medical doctors at different levels: obstetrics and gynecology specialists, post-graduate students, medical officers, house surgeons, and nurses working in maternity care) and facility administrators from the selected facilities. The healthcare providers and facility administrators were contacted by a member of the research team (either KLS, APP, or KTA) at their place of work in the study facilities and invited to participate in the study. Purposive quota sampling was applied to achieve a stratified sample without random sampling, with diversity in terms of the type of healthcare provider, years of experience, and gender.

### Data collection, management, and analysis

The data collection tools included semi-structured discussion guides covering the following themes (S1 Text): availability and source of the variables required for the Robson classification, readiness of facilities and challenges to implementation, responsible person for

implementation, interval period of CS rate review, role of administrators, and benefits and cost. The guides were piloted in one tertiary hospital and one township hospital before data collection. The data collection team (KLS, APP, NOM, and KTA) received two days intensive training. Data collectors were all medical doctors who had experience in qualitative data collection and included both male (KLS, APP, KTA) and female (NOM) interviewers. Interviews were conducted in an independent room with privacy. Data were collected between July and August 2019. Interviews lasted for about one hour. Data sufficiency was discussed among researchers in the field, acknowledging that we sought diversity in perspectives both across cadre of healthcare provider and based on the study sites. Interviews were recorded and field notes were made during interviews, and transcribed verbatim into Myanmar language by the research team members. De-identified transcripts were kept confidential in the password protected computer and accessible only by the investigators.

In addition to interviews, we conducted readiness assessments of facility and medical records systems (S2 Text). The assessment included whether the key variables to classify women into the Robson's 10 groups were included in the woman's medical record, consistency of reporting across all records, responsible person to complete the medical record and, audit and feedback mechanism. This information was collected using a semi-structured data collection form to extract the information from three to four randomly selected medical records in each facility setting.

We used the thematic analysis approach described by Braun and Clark [10]. The local and international research team co-developed the codebook during a two-day intensive analysis workshop. The codebook was developed by the themes emerging inductively from the data; availability and sources of data to be used for the Robson classification, its readiness, healthcare providers' and administrators' willingness to implement, challenges, responsible persons for implementation, role of administrators, interval period of CS rate review, advantages and disadvantages of the implementation of the classification. KLS and NSA coded the data using the Atlas.ti qualitative analysis software (version 7 ATLAS. ti Scientific Software Development GmbH, Berlin) and the outputs were generated according to the thematic codes. The research team interpreted and discussed the study results by the themes upon their perspectives. Data analysis was conducted in Myanmar language and quotations supporting emergent themes were translated in English by a member of the research team for inclusion in the results. We triangulated the qualitative research findings with medical records in each health facility in order to improve validity of the research findings. We did this by comparing what participants reported as available data in the medical records to what was typically recorded in the observed medical records.

## Results

A total of 67 in-depth interviews (IDIs) were conducted and no potential participants approached refused to participate. The sociodemographic characteristics of the study participants are described in Table 1. The mean age of the respondents was 42±10.5 years (min 24 years, max 59 years). A range of health professionals participated including obstetrician-gynecologists (37%), administrators (e.g. superintendent, head of obstetrics, matron-in-charge of maternity ward) (28%), midwives or nurses working in maternity (21%), medical officers (10%), and postgraduate student/house surgeon (3%). Most participants were female (84%) and from tertiary level referral hospitals (76%).

Participants were not familiar with the Robson classification system prior to engagement in this project. Typically, current practice for CS audits centered on feedback of 'clinically interesting cases' (e.g. clinically complicated cases). Therefore, to start the interview, the interviewer

**Table 1. Characteristics of the respondents (n = 67).**

| Characteristics | n (%) |
|---|---:|
| Age (Mean±SD) | 41.9±10.5 years |
| *Age* | |
| 24–35 yrs | 23 (34.3%) |
| 36–50 yrs | 25 (37.3) |
| >50 yrs | 19 (28.4) |
| *Gender* | |
| Female | 56 (83.6) |
| Male | 11 (16.4) |
| *Profession* | |
| Administrator | 19 (28.4) |
| Obstetrician-gynecologist | 25 (37.3) |
| Medical officer | 7 (10.5) |
| Postgraduate student/house surgeon | 2 (3.0) |
| Clinical nurse | 14 (20.9) |
| *Hospital Type* | |
| Tertiary Hospital | 51 (76.1) |
| Township/District Hospital | 16 (23.9) |

first described the Robson classification, necessary variables, and what the classification system is used for. Then, we explored barriers and enablers to implement Robson classification system in public hospitals in Myanmar (Table 2).

## Contextual insights from the assessment of medical records

A total of six variables are required to classify into ten mutually exclusive and totally inclusive Robson groups: parity, previous CS, onset of labor, number of fetuses, gestational age, and fetal lie and presentation. In most of the tertiary hospitals, all the variables required for the Robson classification were consistently recorded in the individual woman's medical record across all records reviewed. Usually, the completion of the medical record was done by a post-graduate student or a medical officer (assistant surgeon) during the patient-history taking. However, incomplete information for Robson classification was observed in two township hospitals. In these hospitals, the variables were not recorded in the medical record because the

**Table 2. Barriers and enablers to implement Robson classification system in public hospitals in Myanmar.**

| Enablers | Barriers |
|---|---|
| • Providers' willingness to implement and to learn about the classification and its possibilities<br>• Providers' willingness to be trained in the Robson classification and in electronic databases<br>• Source of the variables required to classify women into the 10 Robson groups were in place and easy to extract in the tertiary hospitals<br>• Engagement of policy-makers and hospital administrators | • Limited human resources<br>• Overburdened providers who could not add the management of data to their already heavy workloads<br>• Limited infrastructure resources<br>• Lack of electronic medical record system in most of the hospitals<br>• Lack of full understanding of the objectives and use of the classification<br>• Viewed as limited value because of CS audits in tertiary hospitals and not providing information on indication for CS<br>• Lack of a standardized medical record which makes data extraction time consuming |

small numbers of women giving birth these hospitals meant that clinicians could memorize the clinical characteristics and progress of the labour. While current practice included some audit and feedback for specific, clinically-interesting or unusual cases who underwent CS, none of the study hospitals had continuing audit and feedback sessions to monitor CS. Aggregate facility CS rates were retrospectively calculated annually, but the Robson groups were not used, thus limiting the ability to use this information to inform quality improvement.

## Availability and source of the variables for the Robson classification

Participants confirmed that the variables required to classify women into the 10 Robson groups were typically collected during patient-history at arrival to the labor ward and noted down on the paper-based patient hospital record. In some tertiary hospitals, these variables were additionally entered and saved electronically in SPSS or Microsoft Excel for facility-level registry. However, the electronic records were either not complete for all variables or had women missing. In addition, the lack of a structured standardized medical record form would make it time consuming to extract any variable and would potentially act as a barrier for efficiency, feasibility and sustainability. Participants noted that occasionally the source of data for the Robson classification in tertiary hospitals was the duty report (hand-written report to the head of department in the next day), but not all essential variables for the Robson classification were available in this report. Some township hospitals did not collect or note down any of the variables on the hospital record because there were fewer patients and healthcare providers reported that they could memorize their patients' conditions and progress without written notes:

> "These (variables) are already presented in the history taking process and noted down on the paper." (A medical doctor from a tertiary hospital)

> "I bear in mind because there were usually one or two patient(s) only. Sometimes, I noted down (on the chart) such as when the contraction started, how many centimeters of cervical dilatation while sometimes, I did not. I had to memorize the condition of my patients." (A medical doctor from a township hospital)

The primary source of the variables for the classification was the paper-based individual patient record in almost all hospitals while a necessity of standardized format is prominent.

## Provider's readiness and willingness to implement the Robson classification

Most participants believed that collecting the Robson variables would not be difficult if there were sufficient human resources and training, because these variables were usually recorded in the medical records. The respondents also mentioned the requirement of uniform format to be systematic.

> "There is feasibility. It [Robson classification] is good to implement. Even in the township hospital, they could monitor from the data and meanwhile, it is ok for Robson. It should be implemented because it is not a heavy job." (A medical doctor in a tertiary hospital)

Most participants were interested in Robson classification and how it might help them to better understand the patient profile in their hospitals but recommended to pilot first before implementing. However, some participants thought that collecting and preparing the Robson classification would have limited added value, due to existing audits for CS conducted in

clinically-interesting or unusual cases and the fact that the Robson classification does not assess clinical appropriateness or indication of a specific CS. Almost all participants believed that the Robson classification would be helpful to implement in township and station hospitals, where CS audit were not routinely happening:

> "This is really good job. There might be many papers coming out from the data. Main issue is the interest." (A medical doctor in a tertiary hospital)

> "Robson classification only might not be useful for us. We do not want that kind of data. We had the audits for reducing caesarean rate, complication sepsis, abortion-related near misses, PPH (post-partum hemorrhage) may result after PE (pre-eclampsia). That kind of information is more valuable to us." (A medical doctor in a tertiary hospital)

This suggests that there will be an important component of training ahead of implementation of the Robson classification, for healthcare providers to better understand how the data can be used to improve practice, care and ultimately, maternal and perinatal outcomes.

## Challenges in Robson classification implementation

Limited human resources, heavy workloads, and infrastructure resources were the major challenges described that may hinder implementation. Incomplete data was reported to be the result of overburdened medical doctors who have to cover their clinical duties in the hospital as well as data entry. Rotation of duty, lack of staff and of a focal person for data management may affect sustainability and prevent complete electronic data entry. Some participants mentioned the importance of having uniform definitions for the variables of interest across hospitals to ensure consistency and correctness of the data and to enhance meaningful comparisons between hospitals and over time in the same hospital. Finally, infrastructure resources, such as availability of a computer in the hospital, physical space for data management, and limited training for electronic data entry were also mentioned as barriers.

> "There is difficulty with manpower. There are only two nurses in the ward and we have to do injection, tablets for the patients, complaints solving, ward changes and report writing." (A nurse from a tertiary hospital)

> "Many missing in the data and there were different usage of same variable. Data might be entered as "one scar" in this month and "previous one scar" in another month. Furthermore, variable number one in this month was "one scar" while "breech presentation" in another month resulting data could not be combined." (A medical doctor in a tertiary hospital)

This highlights that for reliable implementation of the Robson classification in the study facilities, there are crucial unmet needs of dedicated human resources with sufficient knowledge on medical terminology and technology, and on data entry standardization.

## Responsible people for implementation

Participants noted that the suggested focal person for data collection and entry varied by level of facility. Most mentioned a medical doctor as appropriate, because they understand medical terminology comfortably. In the tertiary level hospital, obstetrician/gynecologist specialist and post-graduate Obstetrics and Gynecology students were viewed as the most suited focal persons. Few medical doctors mentioned nurses as the most appropriate focal person. For the

district hospitals, participants suggested to appoint someone permanently responsible for data collection, and preferably a medical doctor. A data assistant or data manager could be an option if supervised by a medical doctor. Participants in township hospital mostly mentioned nurses as appropriate focal people because of limited medical doctors in these hospitals. However, they also mentioned that most of the nurses could not handle computerized system, which may present challenges if electronic data collection was preferred. They also mentioned the necessity of dedicated person for data collection which would involve extracting the data from the paper-based medical records and doing the data entry into the computer.

For data analysis and reporting, all levels of hospital described medical doctor would be the most suitable and indicated that training would be necessary for data extraction, data analysis and data presentation.

> "I think this is not very difficult. This could be trained and done even by a house surgeon or an outsider. Most important is dedication which is daily or weekly data entry. If there is such dedicated person, it should be ok." (A medical doctor from a tertiary hospital)

> "Medical officer (doctor) is the most suitable (for data analysis and reporting) but training regarding the information to be sought and data analysis is required." (An administrator from a tertiary hospital)

> "It would be better to decide after training. The focal person should fit with the role and responsibility, and good at statistics." (An administrator from a tertiary hospital)

In addition to the person collecting and analyzing the data, the respondents highlighted the important role of policy enforcement for the sustainability of data collection, interpretation and feedback. Other important aspects described were human resource management for data entry, analysis, interpretation, and advocacy to the Obstetrics and Gynecology Department.

> "If we implement this (Robson classification), we need to advocate first to Obstetrics and Gynecology Ward and OPD (Out Patient Department), and medical record team. After that, we need to assign and manage human resource. We also need to encourage data completeness." (An administrator from a tertiary hospital)

> "First, we need to assess whether it is beneficial or not. If yes and there is a policy, we have to request resources. We can implement ourselves, and not very difficult but to be sustainable, we cannot rely upon ourselves." (An administrator from a district hospital)

Policy and supports from the administrators would play a very important role in implementing and sustaining the use of the Robson classification.

## Review interval period

Participants made suggestions about the periodicity of data review ranging from two weeks to annually. Monthly review was the most frequent answer while some suggested CS audits to be conducted fortnightly. Some participants believed that it should depend on the level of hospital, number of patients and data availability.

> "Every 3 months but it might depend on the hospital such as how many patients attended and CS rate. Our hospital does not have so many patients and every 3 months should be ok." (A house surgeon from a tertiary hospital)

"Monthly is the best. Annually could have chance of data drop out and raise issue in data collection. Monthly is the best to analyze and present data. Weekly is better but not easy because of human resource constraint." (A medical doctor from a tertiary hospital)

"It should go as the (CS) audit. It depends on the number of patients too." (An administrator from a tertiary hospital)

"Its shortest interval should take at least every 2 months because I cannot do it alone. It should involve OG (Obstetrics and Gynecology specialist), AS (assistant surgeon/medical officer), nurses, and one or two administrative staff. We have to work hard to handle data around 800 patients for 2 months." (An administrator from a district hospital)

Although the review interval period could therefore differ according to the various factors, agreeing and setting a fixed schedule in each hospital for continuous monitoring of CS rate is perceived as required.

## Perceived advantages and disadvantages of Robson implementation

The participants described some perceived advantages of Robson implementation such as obtaining hospital-based data to monitor CS rates, which, if properly used, analyzed, understood and acted upon, could lead to reducing unnecessary CS. Moreover, implementing the Robson classification could be the first step to raise awareness among policy makers and healthcare professionals regarding the increasing use of CS and the need for action to optimize use.

"I think it can reduce CS rate because there is increasing trend in CS. Not taking proper antenatal care could lead to CS and PE (pre-eclampsia) is another reason. Most of the patients with PE underwent CS. Patients who undergo CS have high risk and I want to reduce CS rate." (A nurse from a tertiary hospital)

"[The Robson classification] is more sensitive tool and can justify the use of CS in our hospital rather than using the overall hospital-level CS rate and just say low rate or high rate of CS." (A medical doctor from a tertiary hospital)

Very few participants expressed disadvantages of Robson implementation. Some respondents described that the extra work required for the implementation of the classification would result in limited benefit because of the existing regular CS audits in tertiary hospitals which present more detailed information and include the indication for CS. One respondent suggested to use the classification in the township or station level hospital because these hospitals lack specialist doctors and CS audits.

"So be it. This could be busier for us. We already have detailed information such as indication, breech, fetal distress. Then, we could categorize the number due to fetal distress, due to previous scar, etc. It just not grouping." (A medical doctor from a tertiary hospital)

"This should apply to SMO/TMO level (Station hospital and township hospital levels). Frankly saying, they do not need (CS) audit and do not have it." (A medical doctor from a tertiary hospital)

This suggests that both advocacy and training are required prior implementation of Robson classification system in the hospital for complete and accurate data, and furthermore, sustainability. It also underlines that clinicians find the information on indication for the CS useful

and important to collect and efforts should consider the inclusion of indications within the Robson classification to fill this gap.

## Discussion

Our study constitutes the first report exploring the feasibility and readiness to implement the Robson classification to monitor the use of CS in public hospitals across Myanmar. We found that positive perceptions, acceptability and willingness to implement the classification exist in the hospitals while challenges are foreseen including limited of human and infrastructure resources, overburdened healthcare providers, and limited knowledge to manage electronic databases. Most healthcare administrators and providers had positive attitudes towards implementation of Robson classification system because women clinical records are routinely implemented and contained the variables required for the Robson classification so no additional workload would be required to collect the data. Hospital-based routine collection of variables is perceived with advantages for generating research questions and to support evidence-based policy making and change in clinical practices including for the optimal use of CS.

This study also assessed readiness by observation checklist. Although all the variables required for Robson classification are recorded in the individual patient record chart, the lack of a structured standardized medical records will make it difficult and time consuming to extract these variables and would potentially act as a barrier for efficiency, feasibility and sustainability. This is similar to a study conducted in Sri Lanka to introduce Robson classification [11], variables were routinely collected using standardized form, entered into a database, and routinely checked the data quality. Furthermore, structured and standardized electronic computerized system could reduce missing data [12,13].

Although Robson classification users in other settings have confirmed its simplicity and robustness [14], human and infrastructure resources have to be in place for successful implementation. Having the knowledge and understanding of medical terminology is necessary to avoid misclassification and keep its comparability within and between health facilities and for continuous monitoring of the use of CS in the facility itself. Limited healthcare workforce was described as substantial barrier to implement Robson classification system in the Myanmar context. As of October 2016, a total of 16,292 medical doctors and 36,054 nurses were working under the Ministry of Health [15]. Doctor population ratio was 0.37 per 1,000 population and 1.47 health workforce per 1,000 population which is far below the WHO minimum recommendation [15,16]. Therefore, healthcare workers including medical doctors and nurses have a significant workload and difficult to spare time for routine data collection and abstraction required for the Robson classification. Health information technologists are a relatively recent under-graduate program in Myanmar. They are trained in the universities of medical technology and they could be an option for routine data collection in the future.

Limited infrastructure was described as one of the challenges to implement the Robson classification system. Public hospitals in Myanmar are using paper-based medical record system which is not long-lasing and are prompt to missing important information or data. Missing data on core variables used to classify women into Robson groups has been described in the literature as an important challenge that can lead to unclassifiable women making the results and the interpretation unreliable [11]. More than half of errors made by medical doctors in other settings have been attributed to poor handwriting [17]. Nowadays, electronic medical record system are recommended for comprehensive information, better information flow, and timely decision making improving the quality of care. However, establishing electronic

medical record system requires technical ability, trained persons, and financial support for its sustainability, and may not yet be feasible in Myanmar.

Audit and feedback using the Robson classification has been suggested to optimize the use of CS and it is recommended by WHO [18,19]. Tertiary hospitals in Myanmar participating in our study were conducting CS audits in case-by-case basis for interesting clinical cases. However, using the Robson classification to monitor and understand the use of CS at hospital level can act as a framework for appropriate and meaningful feedback fostering engagement of healthcare providers towards improved quality of care. In Sri Lanka, Senanayake et. al. used the WHO Robson classification implementation manual to implement Robson classification in a hospital setting, and provides examples for how to use the implementation manual in an action-oriented manner in order to improve quality of care [11].

The role of advocacy and policy development at national and hospital level should not be underestimated, particularly in low-resource settings. Both governments and clinicians have expressed concern about the unprecedented rise of CS rates and the potential negative consequences for maternal and infant health. Particularly in LMIC where inequalities in the use of CS are growing and overuse and underuse coexist, policy-makers should prioritize an effective acquisition of data for monitoring and assessing the use of interventions, practices and outcomes. The Robson classification is a tool that allows a rational and action-oriented understanding of birth practices and outcomes identifying, analyzing and focusing interventions on specific and relevant groups for each maternity.

In addition to the use of the Robson classification, WHO recommends making the results of the classification publicly available [1]. Comparisons between maternities across the world can foster international exchange, data-driven discussions on practices, and learning from each other globally. To facilitate sharing, WHO has launched the Robson Classification Platform [20]. This global, free, interactive platform is a place where individual maternity units worldwide can upload and share their hospital-level Robson classifications allowing monitoring and comparison of CS rates across maternity units and countries, and over time. Particularly in LMICs, the WHO Platform could be a powerful tool not only for healthcare providers but also for policymakers who, for a small investment, can see their countries and maternities benefiting from guidance from world-class experts in maternities with optimal maternal and perinatal outcomes, as well as participating in international debates with real-time data.

In Myanmar, CS rate were calculated annually at hospitals aggregately but not categorized into Robson groups. This study was the formative research prior to implementation of the Robson classification in Myanmar. Based on our findings and with support and funding from Ministry of Health, a pilot implementation study was planned to introduce the Robson classification in a tertiary teaching hospital in Myanmar. Ultimately aiming to implement continuous and regular use across all hospitals in Myanmar.

Our study had strengths and limitations. This is the first study assessing the feasibility and readiness to implement the Robson classification in Myanmar. We triangulated our qualitative findings with the assessment by observing the medical records in the facility. In addition, our study has broad representation in Myanmar, with coverage across multiple regions and states, including hospitals affiliated with all medical schools. While we believe these settings to be a crucial starting point to implement the Robson classification as all medical students are trained in these hospital settings, the downside is that these hospitals are all located in the major cities and urban setting, and thus not represent the primary health care setting. Furthermore, we could not cover private hospitals. However, we believe that these hospitals are the entry point for implementing and establishing the use of the Robson classification system because of its resources compared to other hospitals in Myanmar.

## Conclusion

Most healthcare administrators and healthcare providers in the participating hospitals in Myanmar were willing to implement the Robson classification system to monitor and assess the use of CS in their facilities which had routine collection of the required variables. However, significant challenges were also described including limited infrastructure, technology, human resources, and the available paper-based medical record system in the facilities. To address these challenges and promote sustainable use of Robson classification, it would be necessary to establish a standardized format of the paper-based individual record, to ensure adequate human resources, and dedicated assigned person for data entry, analysis, and interpretation.

## Supporting information

**S1 Checklist.**
(PDF)

**S1 Text. Qualitative interview guide.**
(DOCX)

**S2 Text. Facility assessment form.**
(DOCX)

## Acknowledgments

We would like to express our sincere thanks to Department of Medical Services, Ministry of Health for their kind collaboration and administrative support to the study. We are very grateful to hospital administers and healthcare providers from Obstetrics and Gynecology Department from the included hospitals for their corporation and participation in the study. We thank the interviewers for their enthusiastic and active participation in data collection.

## Author Contributions

**Conceptualization:** Kyaw Lwin Show, Thae Maung Maung, Özge Tunçalp, Khaing Nwe Tin, Meghan A. Bohren.

**Data curation:** Kyaw Lwin Show, Thae Maung Maung, Aung Pyae Phyo, Kyaw Thet Aung, Nyein Su Aye, Nwe Oo Mon.

**Formal analysis:** Kyaw Lwin Show, Thae Maung Maung, Chetta Ngamjarus, Nyein Su Aye, Meghan A. Bohren.

**Funding acquisition:** Thae Maung Maung, Özge Tunçalp, Meghan A. Bohren.

**Methodology:** Kyaw Lwin Show, Thae Maung Maung, Aung Pyae Phyo, Kyaw Thet Aung, Meghan A. Bohren.

**Project administration:** Kyaw Lwin Show, Thae Maung Maung, Aung Pyae Phyo, Kyaw Thet Aung, Saw Kler Ku, Nwe Oo Mon, Meghan A. Bohren.

**Supervision:** Thae Maung Maung, Chetta Ngamjarus, Özge Tunçalp, Ana Pilar Betrán, Pisake Lumbiganon, Meghan A. Bohren.

**Writing – original draft:** Kyaw Lwin Show, Thae Maung Maung, Meghan A. Bohren.

**Writing – review & editing:** Aung Pyae Phyo, Kyaw Thet Aung, Chetta Ngamjarus, Nyein Su Aye, Özge Tunçalp, Ana Pilar Betrán, Saw Kler Ku, Pisake Lumbiganon, Khaing Nwe Tin, Nwe Oo Mon.

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
