## [Decision Letter · Decision Letter 0]

28 Oct 2022

PGPH-D-22-00989

Feasibility and readiness to implement Robson classification to monitor caesarean sections in public hospitals in Myanmar: formative research

Dear Dr. Show,

Thank you for submitting your manuscript to PLOS Global Public Health. After careful consideration, we feel that it has merit but does not fully meet PLOS Global Public Health’s publication criteria as it currently stands. Therefore, we invite you to submit a revised version of the manuscript that addresses the points raised during the review process.

We invite you to submit a revised version of your manuscript, taking into consideration the comments received from the two reviewers. In particular, to improve overall clarity, please add a table summarizing the barriers and enablers of feasibility and readiness to implement the Robson classification system in public hospitals in Myanmar. In addition, please ensure that all items in the COREQ checklist are reported in the manuscript. 

We look forward to receiving your revised manuscript.

Kind regards,

Melissa Morgan Medvedev, M.D., Ph.D.

Academic Editor

Journal Requirements:

2.  Please insert an Ethics Statement at the beginning of your Methods section, under a subheading 'Ethics Statement'. It must include:

a. (for human participants/donors) - A statement that formal consent was obtained (must state whether verbal/written) OR the reason consent was not obtained (e.g. anonymity). NOTE: If child participants, the statement must declare that formal consent was obtained from the parent/guardian.

3.  Please amend your detailed Financial Disclosure statement. This is published with the article. It must therefore be completed in full sentences and contain the exact wording you wish to be published.

Additional Editor Comments (if provided):

Reviewers' comments:

Reviewer's Responses to Questions

**Comments to the Author**

1. Does this manuscript meet PLOS Global Public Health’s publication criteria? Is the manuscript technically sound, and do the data support the conclusions? The manuscript must describe methodologically and ethically rigorous research with conclusions that are appropriately drawn based on the data presented.

Reviewer #1: Yes

Reviewer #2: Yes

2. Has the statistical analysis been performed appropriately and rigorously?

Reviewer #1: N/A

Reviewer #2: N/A

3. Have the authors made all data underlying the findings in their manuscript fully available (please refer to the Data Availability Statement at the start of the manuscript PDF file)?

Reviewer #1: Yes

Reviewer #2: Yes

4. Is the manuscript presented in an intelligible fashion and written in standard English?

Reviewer #1: No

Reviewer #2: Yes

5. Review Comments to the Author

Reviewer #1: This article addresses an important issue for Myanmar´s maternal care: provider’s perceptions about the feasibility and acceptability to implement a standardized classification system in a context of increasing CS rates to help hospitals monitor those rates. Robson classification system would help them understand which groups contribute most to overall caesarean section rates in order to decide clinical practices for improving clinical management and quality of maternal care.

Minor comments

Plos Global Public health submission guidelines asks to include continuous line numbers in the manuscript file

Methods section

Strategy of Analysis: The use of triangulation to strengthen the analysis was not discussed. You mentioned that “We triangulated our qualitative

findings with the assessment by observing the medical records in the facility” (different strategy to study the same problem: different techniques to get the same data)

Did different subjects were involved to answer the same question, different researchers for the same analysis?

Even though you stated that the analysis was reported according to the Consolidated criteria for reporting qualitative research (COREQ) there are many ítems from this checklist that are not reported in the manuscript. My suggestions is to go through the checklist and the manuscript and include the missing information. For instance: Researcher reflexivity was not discussed, I think that publishing some reflexive content would allow readers to sense how the researcher shaped the entire project, and in particular, the interpretation of findings (Discussion of relationship between researcher and participants during fieldwork; Demonstration of researcher’s influence on stages of research process; Evidence of self-awareness/insight; Documentation of effects of the research on researcher; evidence of how problems/complications met were dealt with)

Results section

In this verbatim: “Robson classification only might not be useful for us. We do not want that kind of data. We had the audits for reducing caesarean rate, complication sepsis, near misses (abortion), PPH (post-partum hemorrhage) may result after PE (pre-eclampsia). That kind of information is more valuable to us.” (A medical doctor in a tertiary hospital)

I think there is a mistake in “…near misses (abortion)…” Were you referring to NM due to abortion?

I feel that pieces of information are repeated under the way the analysis was organized, and that also it does not follow a connecting thread. The manuscript would benefit from having a table organizing and conceptualizing the findings that might hinder or enable feasibility, acceptability and readiness to implement the Robson classification system in public hospitals across Myanmar.

Reviewer #2: Dear editor,

thanks for the opportunity of reviewing this article. It seems to me a very good report regarding Robson's Classification, from a qualitative perspective. I think it is the first report using this approach on Robson's Classification. I have some few questions regarding the manuscript, and I believe the manuscript should be accepted for publication.

1) In Methods, authors defined that they would interview professionals from ten different hospitals, 5 tertiary and 5 primary sites of care. However, in results, 76% of participants were from tertiary level referral hospitals. I understand that the sample was not supposed to be representative, according to the study design, however I think authors should justify why they included more professionals from tertiary services.

2) In page 13, I guess the adequate term is "electronically", instead of "electrically"

3) Page 17: is "3 monthly" correct or is it "3 months"?

6. PLOS authors have the option to publish the peer review history of their article (what does this mean?). If published, this will include your full peer review and any attached files.

**Do you want your identity to be public for this peer review?** For information about this choice, including consent withdrawal, please see our Privacy Policy.

Reviewer #1: No

Reviewer #2: **Yes: **Jose Paulo de Siqueira Guida

---

## [Decision Letter · Decision Letter 1]

4 Jan 2023

Feasibility and readiness to implement Robson classification to monitor caesarean sections in public hospitals in Myanmar: formative research

PGPH-D-22-00989R1

Dear Dr. Show,

We are pleased to inform you that your manuscript 'Feasibility and readiness to implement Robson classification to monitor caesarean sections in public hospitals in Myanmar: formative research' has been provisionally accepted for publication in PLOS Global Public Health.

Best regards,

Melissa Morgan Medvedev, M.D., Ph.D.

Academic Editor

Reviewer Comments (if any, and for reference):

Reviewer's Responses to Questions

**Comments to the Author**

1. If the authors have adequately addressed your comments raised in a previous round of review and you feel that this manuscript is now acceptable for publication, you may indicate that here to bypass the “Comments to the Author” section, enter your conflict of interest statement in the “Confidential to Editor” section, and submit your "Accept" recommendation.

Reviewer #1: All comments have been addressed

2. Does this manuscript meet PLOS Global Public Health’s publication criteria? Is the manuscript technically sound, and do the data support the conclusions? The manuscript must describe methodologically and ethically rigorous research with conclusions that are appropriately drawn based on the data presented.

Reviewer #1: Yes

3. Has the statistical analysis been performed appropriately and rigorously?

Reviewer #1: N/A

4. Have the authors made all data underlying the findings in their manuscript fully available (please refer to the Data Availability Statement at the start of the manuscript PDF file)?

Reviewer #1: Yes

5. Is the manuscript presented in an intelligible fashion and written in standard English?

Reviewer #1: Yes

6. Review Comments to the Author

Reviewer #1: This article addresses an important issue for Myanmar´s maternal care: provider’s perceptions

about the feasibility and acceptability to implement a standardized classification system in a

context of increasing CS rates to help hospitals monitor those rates. Thank you for addressing my comments. I have no further suggestions

7. PLOS authors have the option to publish the peer review history of their article (what does this mean?). If published, this will include your full peer review and any attached files.

**Do you want your identity to be public for this peer review?** For information about this choice, including consent withdrawal, please see our Privacy Policy.

Reviewer #1: **Yes: **Mercedes Colomar
